# Oncology Clinic-Based Hereditary Cancer Genetic Testing in a Population-Based Health Care System

**DOI:** 10.3390/cancers12020338

**Published:** 2020-02-03

**Authors:** Matthew Richardson, Hae Jung Min, Quan Hong, Katie Compton, Sze Wing Mung, Zoe Lohn, Jennifer Nuk, Mary McCullum, Cheryl Portigal-Todd, Aly Karsan, Dean Regier, Lori A. Brotto, Sophie Sun, Kasmintan A. Schrader

**Affiliations:** 1Interdisciplinary Oncology Program, Faculty of Medicine, University of British Columbia, Vancouver, BC V6T 1Z3, Canada; matthew.richardson@bccancer.bc.ca; 2Hereditary Cancer Program, BC Cancer, Vancouver, BC V5Z 1K1, Canada; haejungmin.1992@gmail.com (H.J.M.); quan.hong@bccancer.bc.ca (Q.H.); katie.compton1@bccancer.bc.ca (K.C.); SzeWing.Mung@bccancer.bc.ca (S.W.M.); ZLohn@cw.bc.ca (Z.L.); jnuk@bccancer.bc.ca (J.N.); MMcCullum@bccancer.bc.ca (M.M.); CPortigal@bccancer.bc.ca (C.P.-T.); 3Department of Pathology and Laboratory Medicine, University of British Columbia, Vancouver, BC V6T 2B5, Canada; akarsan@bcgsc.ca; 4Department of School of Population and Public Health, Faculty of Medicine, University of British Columbia, Vancouver, BC V6T 1Z3, Canada; dregier@bccrc.ca; 5Cancer Control Research, BC Cancer, Vancouver, BC V5Z 1L3, Canada; 6Department of Obstetrics and Gynecology, Faculty of Medicine, University of British Columbia, Vancouver, BC V6Z 2K8, Canada; lori.brotto@cw.bc.ca; 7Division of Medical Oncology, Faculty of Medicine, University of British Columbia, Vancouver, BC V5Z 1M9, Canada; 8Department of Medicine, University of British Columbia, Vancouver, BC V5Z 1M9, Canada; 9Department of Medical Genetics, University of British Columbia, Vancouver, BC V6H 3N1, Canada; 10Department of Molecular Oncology, BC Cancer, Vancouver, BC V5Z 1G1, Canada

**Keywords:** hereditary cancer, genetic counselling, genetic testing, patient reported outcome measures

## Abstract

New streamlined models for genetic counseling and genetic testing have recently been developed in response to increasing demand for cancer genetic services. To improve access and decrease wait times, we implemented an oncology clinic-based genetic testing model for breast and ovarian cancer patients in a publicly funded population-based health care setting in British Columbia, Canada. This observational study evaluated the oncology clinic-based model as compared to a traditional one-on-one approach with a genetic counsellor using a multi-gene panel testing approach. The primary objectives were to evaluate wait times and patient reported outcome measures between the oncology clinic-based and traditional genetic counselling models. Secondary objectives were to describe oncologist and genetic counsellor acceptability and experience. Wait times from referral to return of genetic testing results were assessed for 400 patients with breast and/or ovarian cancer undergoing genetic testing for hereditary breast and ovarian cancer from June 2015 to August 2017. Patient wait times from referral to return of results were significantly shorter with the oncology clinic-based model as compared to the traditional model (403 vs. 191 days; *p* < 0.001). A subset of 148 patients (traditional *n* = 99; oncology clinic-based *n* = 49) completed study surveys to assess uncertainty, distress, and patient experience. Responses were similar between both models. Healthcare providers survey responses indicated they believed the oncology clinic-based model was acceptable and a positive experience. Oncology clinic-based genetic testing using a multi-gene panel approach and post-test counselling with a genetic counsellor significantly reduced wait times and is acceptable for patients and health care providers.

## 1. Introduction

Emerging use of genetic testing to guide cancer therapies, combined with greater public and health care provider awareness, has led to rising demand for publicly funded cancer genetic services [1,2]. This has resulted in longer wait times for patients to access genetic counselling and testing.

Cancer genetic counselling has traditionally been modeled after services provided for families with Huntington’s disease where intensive one-on-one counselling before and after genetic testing is completed given the potential psychological consequences of a positive result for an untreatable disease [3]. Although genetic testing for common cancer susceptibility genes such as *BRCA1* and *BRCA2* can lead to psychological distress [4], most patients do not currently display increased distress and anxiety [5,6]. Further, results can inform prevention, early detection, and treatment. Knowledge of germline *BRCA1* or *BRCA2* variant status results in increased breast cancer surveillance, earlier breast cancer diagnosis, increased gynecological cancer surveillance, and prevention of fallopian tube and ovarian cancers [7]. Given the rapidly expanding indications for genetic testing to guide oncologic treatment decision-making, alternative genetic counselling models must be explored to ensure timely access to results.

Newer approaches to cancer genetic counselling have recently been described. A process developed in the Netherlands, termed DNA-direct, involving an abbreviated pre-test telephone appointment, supplemented with written and digital information has been described [8]. Patients receiving testing via DNA-direct demonstrated high satisfaction without increased distress when compared to patients undergoing traditional face-to-face pre-test genetic counselling [8]. Local experience at the British Columbia (BC) Hereditary Cancer Program (HCP) has demonstrated that DNA-direct [9], and both videoconference and group counselling models are acceptable to patients [10,11]. 

A streamlined model from the United Kingdom with oncologists providing pre-test counseling and consenting patients for genetic testing has also been described [12]. This study demonstrated that oncology clinic-based genetic testing reduced overall patient wait times for genetic testing services, reduced costs, and was satisfactory for patients [12]. Similarly, an international prospective study in the United States, Italy, and Spain found that ovarian cancer patients were highly satisfied with oncology clinic-based genetic testing for *BRCA1* and *BRCA2* [13]. Similar models using Australian oncologists to provide *BRCA1* and *BRCA2* genetic testing, [14] and Canadian surgeons to provide genetic testing [15] in ovarian cancer patient populations have also been reported.

To improve access to genetic testing, we implemented oncology clinic-based genetic testing using a multi-gene hereditary cancer panel for selected patients with breast and ovarian cancer in British Columbia, Canada. The primary objectives of this study were to evaluate wait times and patient reported outcome measures, while controlling for demographic factors, for the new streamlined oncology clinic-based model as compared to traditional genetic counselling. The secondary objectives of this study were to describe oncologist and genetic counsellor acceptability and experiences. We hypothesized that the oncology clinic-based approach would be feasible, decrease wait times, and not affect patient reported outcome measures. Our study uniquely assesses oncology clinic-based genetic testing and counseling in a publicly funded, population-based healthcare system using multigene panel testing and disclosing all test results to patients. 

## 2. Results

### 2.1. Study Population 

A total of 702 patients that were seen in the HCP from August 2015–July 2017, were invited to take part in this study. Three hundred and twenty two of the 537 patients who received services through the traditional model elected to participate, 78 of the 165 patients who received oncology clinic-based services elected to participate (*n* = 400). Of the 400 consented participants, 259 completed the study survey package. One hundred and forty-eight patients (99 traditional and 49 oncology-clinic based) with a personal history of breast/ovarian cancer met the study inclusion criteria and were included in our evaluation of the patient survey package responses (Figure 1). 

### 2.2. Wait Times

Wait times were calculated from the date of referral to the HCP to the date of return of genetic test results for the 400 individuals seen at the HCP that consented to participate in the study. The mean wait time in days for patients participating in the oncology clinic-based model (M = 191, SD = 174) was significantly shorter (t = 8.05; *p* < 0.001) as compared to the wait time for patients participating in the traditional model (M = 403, SD = 312).

### 2.3. Patient Demographics

Patient demographics for the traditional and oncology clinic-based models are outlined in Table 1. 

The mean age at referral for all patients was 58 years and all patients were female. Patient age, cancer diagnosis, method of testing, test results, and results appointment format are summarized in Table 1. 

For the traditional model, pre-test counselling sessions were in-person for 41.4% (*n* = 41) of patients, and by videoconference or telephone for the remaining. Most post-test results counselling sessions for patients in the traditional model were conducted by telephone (*n* = 91; 91.9%). In comparison, all pre-test sessions were in person at oncology clinics for oncology clinic-based patients and post-test sessions with the genetic counsellor were done by videoconference (*n* = 21; 42.9%), telephone (*n* = 4; 8.2%) or in person (*n* = 24; 42.9%). Patients in the oncology clinic-based model were significantly more likely to receive in-person or videoconference appointment types than traditional model patients (*p* < 0.001).

### 2.4. Genetic Testing

Most patients received genetic testing through the in-house hereditary cancer gene panel through the BC Centre for Clinical Genomics that was expanded from 14 to 17 hereditary cancer predisposition genes in November 2016 (Table 2). Among 148 patients, 55 underwent 14-gene panel testing (37.2%), 77 had 17-gene panel testing (52.0%), and 16 traditional model patients either had only prior *BRCA1* and *BRCA2* testing (*n* = 3; 3.0%) or prior self-funded private-pay testing (*n* = 13; 13.1%). Thirty-two patients (21.6%) met provincial criteria for additional expanded panel testing via send-out to outside laboratories (Including the 13 traditional model patients that only utilized prior private-pay-testing) (Table 1). 

Overall, genetic testing identified a disease causing pathogenic or likely pathogenic variant in 14.2% of patients (*n* = 21) and variant detection rates were comparable across both models (Table 1). *BRCA1* (*n* = 6; 4.1%) and *BRCA2*, (*n* = 6; 4.1%) were the most commonly identified variants. Pathogenic or likely pathogenic variants in monoallelic *MUTYH*, 2.0% (*n* = 3), *PALB2*, 1.4% (*n* = 2), *CHEK2*, 0.7% (*n* = 1), *PTEN*, 0.7% (*n* = 1), *TP53*, 0.7% (*n* = 1), and *APC*, 0.7% (*n* = 1) were also identified. Testing was uninformative for 55.4% of patients (*n* = 82). 23 likely benign variants were identified in 9.5% (*n* = 13) of patients. 42 variants of uncertain significance (VUS) were identified in 20.9% (*n* = 31) of patients. Further, one patient originally reported with a germline pathogenic variant in the *NF1* gene, was found to have it likely as a result of clonal hematopoiesis.

### 2.5. Patient Reported Outcome Measures

For survey analyses, the traditional model (*n* = 99) and oncology clinic-based model (*n* = 49) patients were compared. Patient and genetic testing factors were analyzed to identify any potential confounding factors associated with results from the survey package. The factors analyzed included, knowledge (as measured by the Genetic Knowledge Questionnaire), age at referral, sex, gene and variant status, oncology clinic-based testing criteria met, and family history. Results from the five questionnaires that made up the survey package are summarized in Table 3.

#### 2.5.1. Knowledge Questionnaire

Overall, there were no significant differences in responses between the traditional and oncology clinic-based models (Table 3).

#### 2.5.2. Multidimensional Impact of Cancer Risk Assessment Survey

After adjusting for previously determined associated demographic factors, there were no significant differences in patient distress, uncertainty, or experience scores between the traditional and oncology clinic-based models. 

#### 2.5.3. The Decisional Conflict Scale

After adjusting for previously determined associated demographic factors, there were no significant differences in Decisional Conflict Scale (DCS) scores between the traditional and oncology clinic-based models.

#### 2.5.4. Patient Acceptability Scale

Overall, patients indicated that they were comfortable with the genetic testing process, and that this was acceptable, with no difference between the two models (mean responses 4.54 and 4.53 for the oncology clinic-based model and the traditional model, respectively).

#### 2.5.5. The Genetic Counselling Outcome Scale

The Genetic Counselling Outcome Scale (GCOS-24) was administered after genetic counselling and genetic testing with similar mean scores between the two models (traditional model = 120.93, oncology clinic-based model = 120.17).

### 2.6. Oncologist and Genetic Counsellor Experieince

Among 19 oncologists participating in oncology clinic-based testing, 8 (42%) completed an online survey. Clinician years of practice ranged from 2 to greater than 10 years. The number of oncology clinic-based patients per oncologist ranged from 2–30. An 11-question survey was adapted from George et al. [12] to reflect use of a multi-gene panel. When oncologists were asked if they felt ‘the process for carrying out multi-gene panel testing worked well’, 4 oncologists indicated ‘strongly agree’, 3 indicated ‘agree’, and 1 indicated ‘disagree’. 

Six out of 14 (43%) genetic counsellors completed surveys. The number of oncology clinic-based patients seen per genetic counsellor ranged from 4–65. Genetic counsellors estimated that genetic testing result disclosures for patients undergoing the traditional model on average were 16.7 min (SD = 2.6) (scheduled for 30 min) as compared to 43.3 min (SD = 10.3) (scheduled for 60 min) for the oncology clinic-based patients. Among the 6 genetic counsellors, 4 indicated that their oncology clinic-based patients were ‘usually prepared’, and 2 indicated ‘sometimes prepared’ for their results appointment.

## 3. Discussion

We implemented oncology clinic-based genetic testing using a multi-gene hereditary cancer panel approach in a publicly funded population-based health care setting in British Columbia, Canada. We evaluated wait times, patient reported outcome measures, and oncologist and genetic counsellor experience and compared the streamlined oncology clinic-based model to traditional one-on-one genetic counselling and genetic testing.

The oncology clinic-based model significantly shortened mean wait times from referral to return of genetic test results; 191 days as compared to 403 days for the traditional model. Similarly, other groups have reported reduced time to results with oncology clinic-based *BRCA1* and *BRCA2* testing for ovarian cancer patients [12,13].

Other models including DNA-direct, telephone, videoconference, and group genetic counselling have been reported and these approaches are also acceptable for patients and reduce wait times [8,9,10,11]. An advantage of the oncology clinic-based model is that it can potentially be combined with alternative streamlined approaches such as DNA-Direct to increase efficiency and further decrease wait times.

As compared to previous streamlined studies with *BRCA1* and *BRCA2* testing in ovarian cancer patients only [12], in our study, genetic testing was performed using a multi-gene panel approach in patients with a personal history of breast and/or ovarian cancer. The overall pathogenic variant detection rate was 14.2% (*n* = 21), and comparable to previous reports using multi-gene panel testing in a similar population [16,17]. 

Monoallelic pathogenic variants were identified most commonly in *BRCA1* and *BRCA2* (*n* = 12; 8.1%). As expected with multi-gene panel testing, VUS were common (*n* = 31; 20.9%) and non-*BRCA1* or *BRCA2* pathogenic or likely pathogenic variants comprised 42.9% (*n* = 9) of pathogenic findings. The high proportion of non-*BRCA1* or *BRCA2* pathogenic variants and the discovery of the somatic *NF1* variant likely due to clonal hematopoiesis, highlights the importance of educating clinicians and patients about the broad range of results possible with multi-gene panel testing.

Patients completed five questionnaires to assess various psychosocial outcomes. After controlling for potential confounding demographic factors, there were no differences on the Multidimensional Impact of Cancer Risk Assessment (MICRA) sub-scales in uncertainty, distress, and experience between the two clinical models. The overabundance of zero answers on the Distress and Positive Experiences sub-scales indicated that the majority of patients felt positive and non-distressed about their genetic testing and counselling. Similarly, there was no difference in Decisional Conflict Scale subscale scores between the oncology clinic-based model and traditional model. 

In addition, most patients indicated that they were comfortable with the genetic testing process, and that the oncology clinic-based process was acceptable, with no difference between the two models on the Patient Acceptability Scale.

Overall, oncologists had a favorable experience with a majority indicating that they felt that the oncology clinic-based process worked well. These findings are similar to a previous report by George et al. [12] that showed that oncology clinic-based genetic testing was feasible and favorable among oncologists. Similarly, 80% of oncologists participating in a study in the US and Europe felt that this was an efficient use of their time [13]. Our results support previous studies indicating that oncologists are accepting and willing to offer genetic counselling and initiate genetic testing. Among six genetic counsellors surveyed, most indicated that they felt that patients were prepared for the genetic test results appointment. 

The 30%–40% survey response rates from patients and health care providers in this study are lower than some previously reported studies [13] but are similar to internal survey response rates at BC Cancer. Key limitations to our study include small numbers of oncologist, genetic counsellor, and patient responses, as well as heterogeneity among the patient population and variable genetic testing approaches. Further, due to the small patient sample size it was not possible for us to cluster patients with their specific healthcare provider in our analysis. In addition, measurement of patient empowerment by the GCOS-24 was completed only once, after the genetic counselling results appointment, therefore only descriptive statistics were applied. Similarly, patient genetic knowledge measured by the genetic knowledge questionnaire was only able to be assessed post-genetic counselling appointment and therefore may not address the potential difference in knowledge from the pre-test modalities. The healthcare provider surveys and the patient Acceptability Scale were reviewed by a multidisciplinary team for face validity and therefore only descriptive statistics were applied. As the interaction with a genetic counsellor was common to both models, it remains unclear as to the degree it impacted patient empowerment. This may be relevant when considering oncologist-led genetic testing models that do not include a genetic counsellor-led results appointment.

## 4. Materials and Methods 

### 4.1. Clinical Models

The study was approved by the University of British Columbia Research Ethics Board (H16-02201). Women age ≥19 undergoing index testing, matching provincial Hereditary Breast and Ovarian Cancer (HBOC) testing criteria (Appendix B), and attending the HCP from June 2015 to August 2017 were eligible to participate in the traditional model. The traditional model is the standard genetic counselling process at the HCP and first involved the referral of the patient to the HCP, followed by a one-on-one 45-min pre-test counselling session with a genetic counsellor prior to the patient receiving a requisition form for genetic testing. This was followed by a scheduled 30-min results disclosure appointment approximately one month after the patients’ blood sample was submitted for genetic testing. Pre- and post-test genetic counselling sessions were conducted in person, by videoconference, or by telephone. 

Women age ≥19 attending their regularly scheduled oncology appointment from June 2015 to August 2017 and meeting a subset of provincial HBOC criteria (non-mucinous ovarian cancer, under 35 breast cancer, or under 65 triple-negative breast cancer; Figure A1) were eligible to participate in the oncology clinic-based model. A subset of the provincial HBOC criteria were used to streamline the process for oncologists when identifying patients that qualify for genetic testing. Patients were consented to participate in the new oncology clinic-based model by their oncologist. The oncology clinic-based model involved pre-test counselling, handout of a requisition form for genetic testing, and referral to the HCP by an oncologist in clinic. This was followed by a scheduled 60-min post-test results session with a genetic counsellor conducted either in person, by videoconference, or by telephone approximately one month after the patients’ blood sample was submitted for genetic testing (Figure 2). Patients in the oncology clinic-based model had a longer results session with a genetic counsellor to review both family and personal history that may have impacted the interpretation of the patients’ genetic test results. 

All patients received a results appointment with a genetic counsellor regardless of their genetic test result. Due to British Columbia’s geographical layout and variable patient availability, all patients were given the option for an in person, telephone, or videoconference session. All oncologists were trained by one of the HCP medical directors to provide genetic testing information to patients and consent patients for genetic testing. Oncologists were also provided with reference material including a frequently asked questions (FAQ) information sheet and were provided continued HCP support as needed. Oncologists initially received a standardized script outline and used a standardized patient consent form.

### 4.2. Genetic Testing

Genetic testing was performed using next-generation sequencing of a multi-gene hereditary cancer panel and Multiplex Ligation-Dependent Probe Amplification (MLPA) for *BRCA1* and *BRCA2* on peripheral blood samples. A minority of patients were offered additional expanded panel testing based on clinical presentation and provincial testing guidelines. 

### 4.3. Study Questionnaires

Patients in both models were contacted by a research assistant over telephone approximately two business days after their genetic testing results appointment to consent them for the survey package. Informed consent was obtained, and the survey package was mailed to patients. If patients were unable or unwilling to complete the paper survey package the research assistant offered to complete the survey package with them over the phone. Patients in the traditional model who did not meet the oncology clinic-model’s HBOC subset criteria were excluded from the analysis of the survey package results. These patients were excluded to maintain similar patient eligibility criteria in both models. 

Patients were mailed the survey package and were instructed to complete the package one month after the results appointment date. Patients completed five questionnaires as a part of the survey package: Genetic Counselling Outcome Scale (GCOS-24) (Appendix A) [18], Decisional Conflict Scale (DCS) (Appendix A) [19], Multidimensional Impact of Cancer Risk Assessment (Appendix A) (MICRA) [20], Genetic Knowledge Questionnaire (Appendix A) [21] and Patient Acceptability Scale (Appendix A). Patients were asked to complete their survey package one month after their results appointment to match the procedures in the original validation of MICRA and GCOS.

Oncologists and genetic counsellors were invited to complete surveys to assess feasibility and acceptability of the streamlined model. Oncologists were invited to complete two questionnaires to evaluate their experiences and acceptability: Oncologist Questionnaire (Appendix A) [13] and Oncologist Experience Scale. Genetic counsellors completed a questionnaire to measure their perceptions of patient preparedness, time taken for both models, and to obtain direct feedback (Appendix A).

### 4.4. Statistical Analyses

Data for wait times from referral to HCP to return of genetic test results, indication for genetic testing, method of genetic testing, patient demographics, and personal and family cancer history were obtained through review of the BC Cancer and HCP electronic chart and pedigree databases. 

Descriptive statistics were presented for both categorical and continuous variables. Categorical variables were summarized as frequencies with percentages, and continuous variables were described as means with standard deviations. Fischer’s exact test was used to determine if there were demographic differences in the patient population proportions between the two clinical models. All summarized questionnaire means were calculated using the recommended practices in the original validations of the questionnaires.

An independent sample *t*-test was used to compare wait times for the two clinical models. Generalized linear model analyses were performed to compare psychological outcomes. Missing survey data was replaced using questionnaire specific scoring instructions. The MICRA sub-scales of Distress and Positive Experiences were dichotomized due to an over-abundance of zero answers from patients. All Decisional Conflict Scale sub-scales were transformed into a categorical variable with three levels due to non-normality. To explore associations between patient factors and genetic test results with survey scores, bivariate analyses using linear or logistic models were performed. Multiple linear or logistic regressions were used to examine the impact of the genetic counselling models on survey scores after adjusting for the significant factors identified in the bivariate analysis. A log transformation was applied for continuous outcomes if needed. After adjusting for multiple comparisons using the Bonferroni method, a *p*-value of 0.003 was used to indicate significance for all analyses. Statistical analysis was performed using R V.3.5.1.

## 5. Conclusions

In summary, we conclude that streamlined oncology clinic-based genetic counselling and genetic testing resulted in similar outcomes for patient reported outcome measures, was acceptable to their health care providers, and significantly reduces wait times for genetic testing. Our study is unique as genetic testing was performed in a patient population with different cancer types (breast and/or ovarian cancer) while using a multi-gene panel approach, with all VUS returned to patients, in a publicly funded population-based health care system. 

Efforts to further expand and evaluate the oncology clinic-based model are underway, with recent expansion to include additional oncologists, nurses, nurse practitioners, and general practitioners with oncology expertise. Future directions include implementing surveys online, implementing pre- and post-genetic counselling surveys as part of clinical care to measure empowerment, and to evaluate patient reported outcome measures and clinician acceptability when introducing new models. As well, future directions will increase testing to include hereditary cancer syndromes other than HBOC.

The field of cancer genetic counselling is rapidly evolving due to recent advances in personalized cancer treatments for patients with hereditary cancer. Multidisciplinary development and evaluation of new rapid genetic testing approaches are needed prior to adopting these into routine clinical practice.

## Figures and Tables

**Figure 1 cancers-12-00338-f001:**
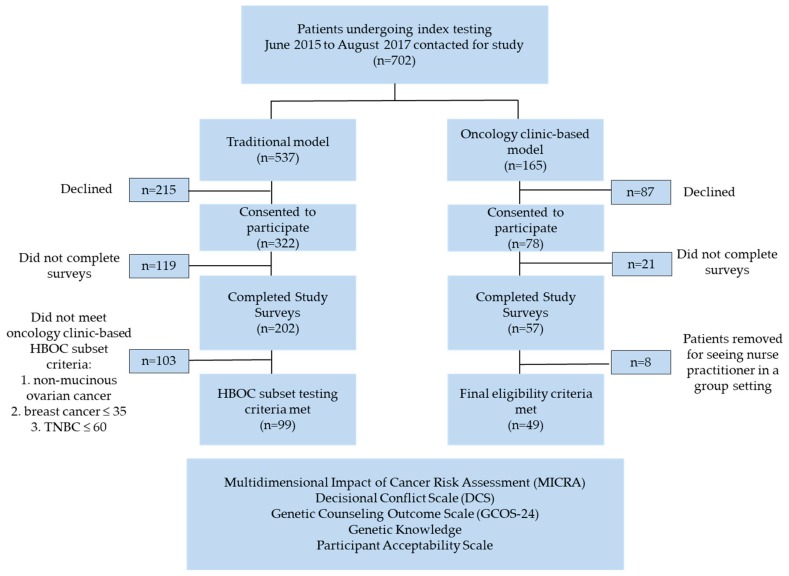
Study Design: The study inclusion criteria with a breakdown of the number of patients included and excluded.

**Figure 2 cancers-12-00338-f002:**
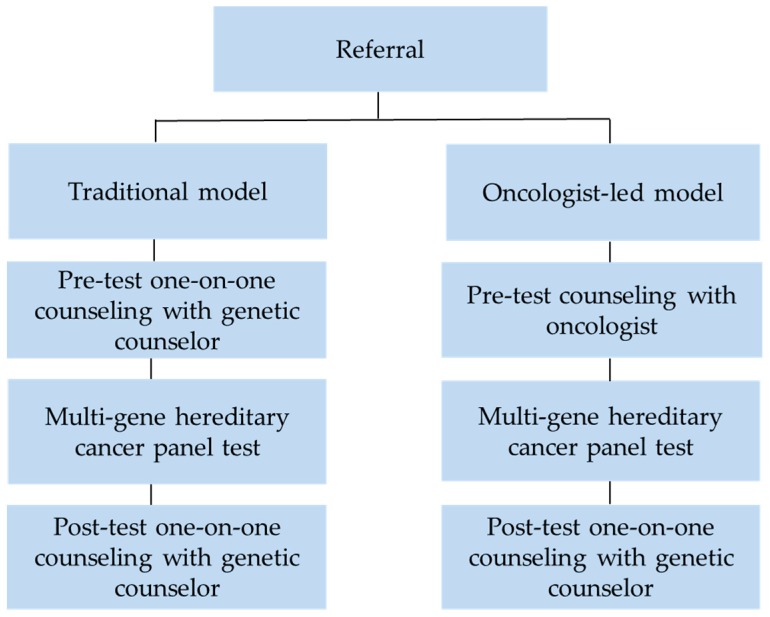
Clinical Models: The traditional clinical model involved pre-test and post-test one-on-one genetic counselling with a genetic counsellor. The streamlined oncology clinic-based model involved pre-test counselling by an oncologist with the initiation of genetic testing at a clinic visit. Once available, patients in both models were provided test results by a genetic counsellor.

**Table 1 cancers-12-00338-t001:** Patient Demographics, Wait Times, Genetic Testing.

	Total Patient Population	Oncology Clinic-Based	Traditional	*p*-Values
**Wait Time** (days)				*p* < 0.001 *
Referral to return of genetic test results (SD)		191 (174)	403 (312)	
**Age** (years, SD)		62.2 (12.0)	56.7 (12.0)	*p* = 0.013
**Gender** (*n,* %)	148 (100)			N/A
Female		49 (100)	99 (100)	
**Personal History** (*n*, %)				*p* = 0.006
Ovarian cancer Breast cancer ≤ 35 Triple negative breast cancer ≤ 60 Breast ≤ 35 and ovarian cancer	86 (58.1) 15 (10.1) 46 (31.1) 1 (0.7)	38 (80.7) 3 (5.3) 8 (14) 0	48 (48.0) 12 (13.0 38 (38.0) 1 (1.0)	
**Family History of Breast/Ovarian Cancer** (*n*, %)				*p* = 1
No Yes	89 (60.1) 59 (39.9)	29 (59.2) 20 (40.8)	60 (60.6) 39 (39.4)	
**Pre-Test Appointment Type** (*n*, %)				*p* < 0.001 *
In-person Telephone Videoconference	90 (60.8) 24 (16.2) 34 (23.0)	49 (100) 0 0	41 (41.4) 24 (24.2) 34 (34.3)	
**Results Appointment Type***(n*, %)				*p* < 0.001 *
In-person Telephone Videoconference	32 (21.6) 95 (64.2) 21 (14.2)	24 (48.9) 4 (8.2) 21 (42.9)	8 (8.1) 91 (91.9) 0	
**Genetic Testing** (*n*, %)				*p* = 0.015
14-gene panel 17-gene panel Prior *BRCA1* and *BRCA2* uninformative Other	55 (37.2) 77 (52.0) 3 (2.0) 13 (8.8)	21 (42.9) 28 (57.1) 0 0	34 (34.3) 49 (49.5) 3 (3.0) 13 (13.1)	
**Additional Expanded Panel Testing** (*n*, %)				*p* = 0.086
No Yes	116 (78.4) 32 (21.6)	43 (87.78) 6 (12.2)	73 (73.7) 26 (26.3)	
**Genetic Test Results ^1^**(*n*, %)				*p* = 0.470
Pathogenic or likely pathogenic Pathogenic monoallelic *MUTYH *^2^ Variant of uncertain significance Likely benign Uninformative	18 (12.2) 3 (2.0) 31 (20.9) 14 (9.5) 82 (55.4)	6 (12.2) 1 (2.0) 12 (24.5) 7 (14.3) 23 (46.9)	12 (12.1) 2 (2.0) 19 (19.2) 7 (7.1) 59 (59.6)	
**Pathogenic *BRCA* Variant **(*n*, %)				*p* = 0.507
No Yes	136 (91.9) 12 (8.1)	44 (89.8) 5 (10.2)	92 (92.9) 7 (7.1)	

^1^ Patients are reported based on their most clinically relevant variant diagnosis (if diagnosed with both a variant of uncertain significance (VUS) and pathogenic variant then the patient is categorized in ‘Pathogenic or likely pathogenic’). ^2^ One patient was diagnosed with a pathogenic *BRCA1* variant and a pathogenic monoallelic *MUTYH* variant and the patient is classified in the “Pathogenic or likely pathogenic” row. * Indicates significance. A significance level of 0.003 was used to account for multiple comparisons. Patient demographics are for the 148 patients that completed survey packages. For the purpose of this study, an uninformative test result was defined as a negative test result for pathogenic or likely pathogenic variants in an individual who had index genetic testing and patients with VUS were considered separately.

**Table 2 cancers-12-00338-t002:** Hereditary Cancer Program 17-gene Panel.

Gene	Syndrome
*BRCA1, BRCA2*	Hereditary breast and ovarian cancer
*PALB2 **	Hereditary breast and pancreatic cancer
*TP53*	Li Fraumeni syndrome
*PTEN*	*PTEN* hamartoma tumor syndrome (Cowden Syndrome)
*CDH1*	Hereditary diffuse gastric and lobular breast cancer
*MLH1, MSH2, MSH6, PMS2*	Lynch syndrome
*MUTYH*	*MUTYH*-associated polyposis (MAP)
*APC*	Familial adenomatous polyposis (FAP)
*POLE **	Hereditary colorectal cancer and colonic polyposis
*POLD1 **	Hereditary colorectal and uterine cancer; colonic polyposis
*STK11*	Peutz–Jeghers syndrome
*SMAD4, BMPR1A*	Juvenile polyposis syndrome

* The 14-gene panel includes high-penetrant genes: *BRCA1*, *BRCA2*, *TP53*, *PTEN*, *CDH1*, *STK11*, *MLH1*, *MSH2*, *MSH6*, *PMS2*, *MUTYH*, *APC*, *SMAD4*, and *BMPR1A* with multiplex ligation-dependent probe amplification of *BRCA1* and *BRCA2. PALB2*, *POLD1*, and *POLE* were added in November 2016 to create the 17-gene panel.

**Table 3 cancers-12-00338-t003:** Survey Results.

Survey	Oncology Clinic-Based Model		Traditional Model		Population Total	
	*n*	Mean (SD)	*n*	Mean (SD)	*n*	Mean (SD)
Genetic Knowledge	49	8.46 (1.79)	99	8.67 (1.52)	148	8.60 (1.59)
Patient Acceptability Scale	49	4.54 (0.71)	92	4.52 (0.69)	141	4.53 (0.70)
Decision Conflict Scale						
Uncertainty Informed Values Clarity	48 48 48	22.57 (19.52) 19.71 (14.04) 24.13 (17.04)	98 97 97	23.36 (21.25) 18.04 (17.38) 24.22 (19.73)	146 145 145	23.10 (20.63) 18.59 (16.32) 24.19 (18.82)
Support	48	25.18 (18.23)	97	26.61 (20.94)	145	26.13 (20.03)
Effective Decision	48	13.16 (14.32)	97	15.21 (19.43)	145	14.53 (17.88)
Multidimensional Impact of Cancer Risk Assessment						
Distress	49	4.53 (5.65)	99	3.37 (5.24)	148	3.75 (5.39)
Uncertainty	49	9.51 (8.19)	99	10.02 (6.88)	148	9.85 (7.32)
Positive experience	49	6.00 (5.78)	99	4.45 (4.66)	148	4.96 (5.09)
Genetic Counselling Outcome Scale	49	120.17 (16.78)	98	120.93 (15.15)	147	120.67 (15.66)

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
