# Peer review of "Oncology Clinic-Based Hereditary Cancer Genetic Testing in a Population-Based Health Care System"

_cancers, 2020, doi:10.3390/cancers12020338_

Round 1
Reviewer 1 Report
The original paper entitled "Oncology clinic-based hereditary cancer genetic testing in a population-based health care system" is well written. In this study, the authors compared the oncology clinic-based model (with pre-test counseling by an oncologist) to the traditional model (with pre-test counseling by a genetic counselor). The results bring irrelevant knowledge. In addition, there is no explanation regarding the methodology related to the survey, and the rules for calculating the Mean were not explained (Table 3. Survey Result). Genetic Testing Results were presented as if they were to have a relationship with Models. No information on what genes 14-gene panel contains.
Reviewer 2 Report
This study examines the novel workflow of oncologist ordering genetic testing (compared to the traditional model of pre-test genetic counselling) in British Columbia, Canada. This replicates other models seen in the UK, but the first to be conducted in North America. The authors examine many aspects of this initiative including clinical efficiencies and patient perspectives in knowledge. There is some provider perspectives but minimal response rates.
It is a sound study with appropriate design and could be applied to other jurisdictions and genetic conditions. A few minor comments.
The methodology has numerous typos please correct.
Was EPCAM not included in the panel? Why not?
Why were all the patients seen to have disclosure? Was that necessary? The UK Model has all negative patients and VUS disclosed by letter. This is another area of efficiency.
Genes should be italicized throughout.
Reviewer 3 Report
This manuscript compares two models of providing cancer genetic counseling and testing services, a traditional model and an oncology-clinic based model. The introduction describes the importance of utilizing alternative service delivery models and evaluating such models in terms of patient outcomes. In terms of methods, the use of multiple validated measures to evaluate patient outcomes is a strength. The results section is comprehensive and the conclusions accurately reflect the results. Overall this is an interesting, well-executed study. However, some revision is needed to enhance the quality of the reporting. Below are my specific suggestions.
1. Introduction:
1a. For reference 14 and 15, you discuss that they report on use of alternative service delivery models but do not report on outcomes measured. If there are no outcomes, I suggest combining the two into one sentence-Similar models using Australian oncologists to provide BRCA1/2 genetic testing (14)and Canadian surgeons to provide genetic testing (15) in ovarian cancer patient populations have also been reported.
1b. You state your primary objectives are to evaluate wait times and patient reported outcomes for the oncology clinic in comparison to the traditional model. But in the manuscript you also report the association of demographic variables with outcomes. So maybe add "controlling for demographic variables". For the second objective, given the small sample size, it seems more appropriate to say that your objective is to describe (rather than evaluate) oncologist and GC acceptability and experience (and feasibility?). For your hypothesis, With regard to your aims, I think it would be helpful to state how your study is going to add to what is currently know about using oncology-based clinics. What gap will your study fill?
2. Results
2a. Study population. It would be clearer to say "A total of 702 patients were seen in the HCP from August 2015-July 2017, all of whom were invited to take part in this study. Three hundred and twenty two of the 537 women who received services through the traditional model, 78 of the 165 who received oncology clinic-based services elected to participate (n=400). Of the 400 consented participants, 259 completed the study questionnaire. I think the flow diagram is great but thought the wording in the body of the results was a little confusing.
2b. This is both a results and methods reporting suggestion. You alternatively talk about surveys/questionnaires (plural). Did participants get multiple surveys or one survey with multiple measures included in the survey? Please specify and use the same wording to refer to the survey(s) consistently.
2c. Wait-times. Rather than include the "n" in the sentence describing wait times, I would just include it in Table 1. Reporting the "n" in the text distracts from the important information- the wait times. I am not sure there is value in reporting the wait times in those who completed the surveys unless the purpose is to show that non-responders were no different than responders (which would require comparing wait times for survey responders versus nonresponders).
2d. It would be helpful to see p values in the tables as well as "Ns". Are the demographics for the entire population that consented or just those that responded to the survey?
2e. Table 1. I think it would be better to have three columns- one for the results for the whole population, then the two different service delivery models. (Same for Table 3). It would reduce the number of results written in the text and allow visual comparison. In terms of genetic test results, what do you mean by uninformative? To me, uninformative means a negative result (no mutations identified) in a person unaffected with cancer whose affected relative(s) have not had testing. But I think you are using the word uninformative to mean a normal result. Please clarify.
2f. Table 2. This table lists 16 genes but you talk about a 14- and 17 gene panel. Maybe list all 17 genes and then highlight which ones were added when the panel expanded to 17 rather than listing the 17 separately.
2g. Genetic testing results: I don't think Figure 2 is necessary. Plus why are there 24 genes listed rather than 14 or 17- are these for the few that had the expanded testing?
2h. This manuscript contains a large number of statistical analyses. Did you do any sort of correction for multiple analyses (e.g., Bonferroni correction)? If not, please justify.
2i. Overall, for all statistical tests, make sure you include the reporting conventions. For instance, for t-test, report t statistic and p value, not just p value.
2j. Oncologist and GC experience. The estimates of time spent in session seem pretty specific given that they are just estimates. Are they averages across respondents? If so, what is the standard deviation?
3. Discussion:
3a. With regard to the paragraph on mono-allelic pathogenic variants, you say BRCA was identified most often (n=12, 8.1%) but then you state the nonBRCA pathogenic variants comprised 42.9%, but the n=9. How can an "n" of 9 result in a bigger percentage than an "n" of 12?
3b. In the paragraph on MICRA and Decisional Conflict, you discuss factors associated with higher distress/higher uncertainty. But this appears to be for the whole population, not an outcome difference related to model of service delivery used. Reporting these result does not seem to be in line with your stated objectives of comparing outcomes between models. As such, I don't think they add value to the discussion.
4. Methods
4a. What do you mean by genetic testing referral?
4b. Clinic models section. I find the term "survey analysis" confusing. In terms of being consented for survey and wait-time components of the study, were they considered two different studies with different consents or part of one study? The way it is written is confusing. For the traditional patients, when you state patients were contacted by phone two days after their results, was that when they were asked to participate/consented or were they consented? I think it is just the consent (much clearer the way it is written in the study questionnaires section). I suggest that you leave the description of consenting out of the clinic models section since it is in the questionnaires section.
4c. Study questionnaires. As mentioned previously, when you use "study questionnaires" it implies to me that you sent participants multiple surveys. If instead there was one survey that included multiple demographic questions and measures, I suggest using the singular to refer to the survey.
4d. Statistical analyses. Line 344, what are DCS subscales (Decisional conflict scale subscales?) Is transforming the DCS subscales into a categorical variable with three levels in line with how the results of the measure can be evaluated? Although there are multiple analyses, a p value of 0.05 was used to determine significance. Please justify why there is no correction for multiple analyses.
5. Conclusions. Rather than say that using the oncology clinic model did not affect outcome measures, perhaps it is better to say that it resulted in similar outcomes to the traditional model.
Thank you for the opportunity to review this interesting manuscript on a topic of importance to the cancer genetic counseling/testing community.
